# Current Evidence and Limitation of Biomarkers for Detecting Sepsis and Systemic Infection

**DOI:** 10.3390/biomedicines8110494

**Published:** 2020-11-12

**Authors:** Shang-Kai Hung, Hao-Min Lan, Shih-Tsung Han, Chin-Chieh Wu, Kuan-Fu Chen

**Affiliations:** 1Department of Emergency Medicine, Chang Gung Memorial Hospital, Linkou 333, Taiwan; mm200411800@cgmh.org.tw (S.-K.H.); onionslife@adm.cgmh.org.tw (S.-T.H.); 2Department of Education, Chang Gung Memorial Hospital, Kaohsiung 833, Taiwan; kaido2019@cgmh.org.tw; 3Clinical Informatics and Medical Statistics Research Center, Chang Gung University, Taoyuan 333, Taiwan; wujinja@cgmh.org.tw; 4Department of Emergency Medicine, Chang Gung Memorial Hospital, Keelung 204, Taiwan

**Keywords:** sepsis, biomarker, C-reactive protein, procalcitonin, presepsin, interleukin-6, CD64, sTREM-1

## Abstract

Sepsis was recently redefined as a life-threatening disease involving organ dysfunction caused by a dysregulated host response to infection. Biomarkers play an important role in early detection, diagnosis, and prognostication. We reviewed six promising biomarkers for detecting sepsis and systemic infection, including C-reactive protein (CRP), procalcitonin (PCT), interleukin-6 (IL-6), CD64, presepsin, and sTREM-1. Among the recent studies, we found the following risks of bias: only a few studies adopted the random or consecutive sampling strategy; extensive case-control analysis, which worsened the over-estimated performance; most of the studies used post hoc cutoff values; and heterogeneity with respect to the inclusion criteria, small sample sizes, and different quantitative synthesis methods applied in meta-analyses. We recommend that CD64 and presepsin should be considered as the most promising biomarkers for diagnosing sepsis. Future studies should enroll a larger sample size with a cohort rather than a case-control study design. A random or consecutive study design with a pre-specified laboratory threshold, consistent sampling timing, and an updated definition of sepsis will also increase the reliability of the studies. Further investigations of appropriate specimens, testing assays, and cutoff levels for specific biomarkers are also warranted.

## 1. Introduction

Sepsis has been defined as the presence or suspected infection along with systemic inflammatory response syndrome (SIRS) since 1992 [1]. The conventional definition was abandoned in 2016, and “sepsis 3-0” replaced the previous “severe sepsis” to increase the predictive accuracy [2]. Since then, many works have been done by an international task force to evaluate the performance of these new sepsis criteria [3,4]. The incidence of sepsis (sepsis-3 or previously severe sepsis) has not changed significantly since the last decade in Taiwan (Figure 1) [5]. The incidence was estimated around 6 cases per 100 adult hospitalizations in the U.S. [6]. In Asia, 50 sepsis cases were reported per 1000 person-visits in emergency departments (EDs) annually in tertiary healthcare centers [7], and case-fatality rates can be as high as 5–28% for sepsis, and 20–65% for severe sepsis [8]. The diagnosis of sepsis, however, is still not straightforward for the frontline health care providers encountering suspected sepsis patients daily. Blood culture sampling often yields false-negative results while the clinical signs of infection are often unspecific. With cumulative efforts of modern practices, still many patients who fulfilled SIRS criteria but have weak evidence of infection are unnecessarily treated with antimicrobial agents, yet inadequate treatment as a result of delayed diagnosis continues to affect approximately one-fourth of sepsis patients [9]. Because of discrete definitions of sepsis and the complexity of its pathophysiology, there is no single best test for diagnosing sepsis, but promising biomarkers are emerging and are under investigation.

Biomarkers are defined as “characteristics that are objectively measured and evaluated as an indicator of normal biological processes, pathogenic processes, or pharmacologic responses to a therapeutic intervention“, and are commonly used to differentiate between distinct pathogenic conditions, indicate disease severity, guide treatments, monitor therapeutic responses, and predict prognosis [10,11]. However, owing to the technical issues or insufficient evidence, many diagnostic biomarkers for sepsis have been proposed, but only a few are used in the clinical setting [12]. In this article, we will review six promising biomarkers aimed at differentiating between adult patients with sepsis and those without sepsis, as well as the reason why many studies are flawed and prone to biases. We will first introduce the categories of biomarkers, describe the mechanisms and performance of six promising biomarkers, discuss the risk of biases of the current body of literature, and lastly give perspectives for further directions.

## 2. Categories of Biomarkers

According to pathophysiology, the biomarkers of sepsis can be classified into the following seven categories: (1) acute phase reactants, e.g., C-reactive protein (CRP), erythrocyte sedimentation rate, and procalcitonin; (2) proinflammatory cytokines, e.g., interleukin, tumor necrosis factor (TNF), and monocyte chemoattractant protein; (3) biomarkers of activated neutrophils and monocytes, e.g., cluster of differentiation (CD), presepsin, and receptor for advanced glycation end products; (4) infectious organisms and related protein, e.g., high-mobility group box 1 and myeloid-related protein; (5) receptors, e.g., toll-like receptors, TNF receptors, triggering receptor expressed on myeloid cell 1 (TREM-1); (6) anti-inflammatory markers, e.g., monocyte human leukocyte antigen-DR expression, cytotoxic T-lymphocyte-associated protein 4; and (7) biomarkers for organ dysfunction, e.g., liver function test, coagulation, and renal function [13].

## 3. Six Promising Biomarkers

### 3.1. C-Reactive Protein

C-reactive protein (CRP), an acute inflammatory phase protein produced by the liver, is one of the oldest biomarkers, discovered in 1930 by Tillet and Francis. In the first observational study of 108 patients in 1987, Mustard et al. found serial CRP measurement could be used to predict 14-day postoperative septic complications [14]. As a systematic review and meta-analysis in 2016 including 45 studies and a total of 5654 patients indicated, the diagnostic accuracy of CRP for distinguishing patients with sepsis from those with non-infectious SIRS revealed a fair sensitivity of 0.75 (95% confidence interval [CI], 0.69–0.79), specificity of 0.67 (95% CI, 0.58–0.74), and area under the curve (AUC) of 0.77 (95% CI, 0.73–0.81) [15]. However, the reference standard for sepsis in the studies enrolled in this meta-analysis varies. Most studies adopted the ACCP/SCCM (1992) clinical definition. However, some studies used a microbiological definition, which indicates positive culture results. A study by Rishi S Nannan Panday et al. in 2019 demonstrated that culture-positive sepsis is associated with a higher mortality rate than culture-negative sepsis. Thus, a microbiological definition of sepsis may represent a more severe subgroup compared with a clinical definition of sepsis [16]. By including different severities of patient populations, the heterogeneity of the included studies prevents a valid quantitative synthesis of the performance comparison. Another systematic review and meta-analysis updated in 2018 including nine studies comparing the diagnostic accuracy between procalcitonin and CRP for sepsis revealed a similar sensitivity of these two biomarkers (CRP: 0.80, 95% CI: 0.63–0.90, procalcitonin: 0.80, 95% CI: 0.69–0.87) but significantly lower specificity for CRP at 0.61 (95% CI: 0.50–0.72) than procalcitonin at 0.77 (95% CI: 0.60–0.88) [17]. However, we also observed similar high heterogeneity among the studies selected in this review, including different septic populations and sepsis stages. The study enrolled sepsis patients in both the intensive care unit (ICU) and ED with different patient spectra, for example, neutropenic sepsis patients in one study and alcoholic hepatitis patients with sepsis in another. The stages of sepsis were also different, ranging from sepsis, severe sepsis, to septic shock. Previous studies have already found that CRP levels are significantly higher in sepsis patients across the different clinical severity groups, thus further investigations with different subgroup analysis or cutoff levels in different severity groups are warranted [18].

Although both meta-analyses demonstrated that CRP has a moderate degree of sensitivity, the specificity was barely satisfactory. There are many causes for elevated CRP levels other than sepsis, including inflammation, burn injuries, cardiovascular disease, and malignancy, which all contribute to the low specificity and limited utility of CRP as a sepsis biomarker [19,20]. In addition, studies have revealed that patients with decompensated or advanced liver cirrhosis have higher basal CRP levels than non-cirrhotic patients due to chronic hepatic inflammation. However, cirrhotic patients show a reduced increase in their CRP levels during infection compared with noncirrhotic patients. Relatively, some evidence exists to support that CRP would still be elevated among patients with a higher risk of mortality [21]. In one of our network meta-analyses comparing seven biomarkers simultaneously, CRP was found to have significantly higher specificity among patients admitted in ICU (OR 1.65, 95% CI 1.03 to 2.66), while significantly lower specificities were associated with sponsorship for CRP (OR 0.51, 95% CI 0.27 to 0.96) [22]. Accordingly, it would still be likely to find a subgroup where CRP would be predictive of sepsis or the related complications.

### 3.2. Procalcitonin

Discovered in the 1970s, procalcitonin (PCT) is a precursor of calcitonin produced by C-cells of the thyroid gland and is associated with severe bacterial infection [23]. In the past two decades, procalcitonin has become the most widely studied biomarker with respect to sepsis. In a systematic review and meta-analysis with 30 observational studies and a total of 3244 patients, Wacker et al. found a moderate pooled sensitivity of 0.77 and specificity of 0.79 (95% CI, 0.72–0.81 and 0.74–0.81) for procalcitonin as a diagnostic marker of sepsis in critically ill patients and concluded that the results of the procalcitonin test should be interpreted in the context of clinical presentations to facilitate clinical decision-making [24]. Some limitations in Wacker’s study include the high heterogeneity between included studies, lack of a gold standard for sepsis, and publication bias. The cutoff value between enrolled studies also varies (median 1·1 ng/mL, IQR 0.5–2.0), and the above biases hamper the final conclusion. After the emergence of the new sepsis definition, Sepsis-3, established in 2016, many studies re-evaluated the performance of procalcitonin. A retrospective cohort study based on Sepsis-3 showed a sensitivity of 74.8% and a specificity of 63.8% for procalcitonin with respect to diagnosing sepsis in emergency patients and concluded that procalcitonin is a reliable biomarker for detecting sepsis [25]. However, in another cohort study on 157 patients investigating procalcitonin at admission, the authors were unable to discriminate between microbiologically proven and non-proven sepsis in Sepsis-3 criteria-positive critically ill patients with an area under the receiver operating characteristics curve (AUC) of 0.55 (95% CI, 0.46–0.64) [26]. These conflicting results may again have resulted from the wide patient spectrum. As indicated by Figure 2, the studies tend to have a more different distribution of the biomarkers; therefore, they may overestimate the performance of the biomarkers. Further studies involving a more generalizable patient spectrum, an optimal cutoff, and evaluation for performance with the new definition are needed. 

Notably, despite lacking definitive evidence to support the use of procalcitonin as a biomarker of sepsis, established evidence supports its power to assist in managing sepsis patients. Some researchers have proposed that imperfect biomarkers can improve patient outcomes as long as clinical practice is guided in specific clinical scenarios. The systematic review and meta-analysis of Kopterides et al. composing 1311 ICU patients supported that procalcitonin-guided antibiotic therapy could reduce antibiotic exposure of sepsis patients [27]. The systematic review and meta-analysis conducted by Wirz et al. comprising 4482 patients also supported that procalcitonin-guided antibiotic therapy reduces antibiotic exposure of upper respiratory tract infection patients [28].

### 3.3. Interleukin-6

Interleukin-6 (IL-6) is a pleiotropic cytokine discovered in 1986, which is involved in a wide range of pathophysiological activities, including temperature regulation and systemic reaction against inflammatory stimuli. IL-6 is known to induce gene expression and release of CRP from the liver in response to inflammation or infection. The level of IL-6 usually increases earlier than that of PCT and CRP, making it a potential biomarker for early detection among sepsis patients [29,30,31,32]. In 2016, Ma et al. conducted a meta-analysis involving 22 studies with a total of 2680 patients in emergency departments, ordinary wards, and ICUs, with a pooled sensitivity of 0.68 (95% CI, 0.65–0.70), specificity of 0.73 (95% CI, 0.71–0.76), and summary receiver operating characteristic curve (SROC) of 0.80 [33]. The authors concluded that IL-6 could aid in confirming, rather than excluding, sepsis, owing to the relatively high specificity. They also performed a comparative meta-analysis and found that PCT had the best discriminative power to differentiate sepsis from noninfectious SIRS (AUC of PCT: 0.83, IL-6: 0.80, CRP: 0.71). In 2019, another meta-analysis investigating IL-6 to differentiate between infection and non-infection in critically ill patients included six studies with 527 ICU patients, and reported a pooled sensitivity and specificity of 0.73 (95% CI, 0.61–0.82) and 0.76 (95% CI, 0.61–0.87), respectively [34]. Among the subgroup analysis in this meta-analysis, they found IL-6 had the highest diagnostic value among PCT, presepsin, and IL-6 in critically ill patients with organ dysfunction. These two meta-analyses, however, showed significant heterogeneity with respect to the included studies, including admission category, reference standards, sampling time, biomarker assays, and methods. The role of IL-6 in detecting sepsis, therefore, remains inconclusive.

A prospective controlled study in 2019 enrolled 142 subjects (51 with sepsis, 46 with septic shock, and 45 controls) and investigated the diagnostic value of IL-6 according to the Sepsis-3 definition. The study revealed that IL-6 could be used to discriminate the sepsis from the control group and could also distinguish septic shock from sepsis. IL-6 also had a superior diagnostic value compared with PCT and CRP levels. The authors concluded that IL-6 is a valuable biomarker for diagnosing sepsis in accordance with the latest Sepsis-3 definition [35]. However, their control group consisted of patients who met the initial SIRS criteria but did not have “sepsis” recorded at routine radiology, blood and urine, vital sign, or medical history checks in the emergency department. Thus, we find this definition questionable, and we have difficulty in defining the study as a cohort study or a case-control study. Accordingly, we caution readers to wait for more evidence to become available before they apply the findings to their daily clinical practice.

### 3.4. CD64

CD64 is a high-affinity immunoglobulin Fc γ receptor expressed on monocytes, eosinophils, and neutrophils. In resting neutrophils, CD64 expression is very low, and it is significantly increased after activation of the cells in response to infection or exposure to endotoxins within a few hours [36,37]. Thus, CD64 has been proposed as a biomarker for diagnosing sepsis. In 2015, Wang et al. conducted a meta-analysis of CD64 expression on neutrophils as a diagnostic marker of sepsis in critical adult patients in 8 studies involving 1986 patients and found that the pooled sensitivity and specificity were 0.76 (95% CI, 0.73–0.78) and 0.85 (95% CI, 0.82–0.87), respectively [38]. They concluded that neutrophil CD64 is a helpful marker for early diagnosis of sepsis in critically ill patients. We also conducted an updated meta-analysis in 2019 with 14 studies and 2471 patients comparing the accuracy of neutrophil CD64, procalcitonin, and CRP in detecting adult patients with sepsis [39]. The pooled sensitivity and specificity of neutrophil CD64 were 0.87 (95% CI, 0.80–0.92) and 0.89 (95% CI 0.82–0.93), which support the idea that CD64 might be a better diagnostic tool for sepsis then procalcitonin and CRP. Furthermore, in our ongoing network meta-analysis, we found that CD64 displays a comparable sensitivity of 0.87 and superior specificity of 0.99 (95% CI: 0.81.0.92 and 0.92–1.00), among seven biomarkers (procalcitonin, CRP, IL-6, presepsin, CD64, sTREM-1, and lipopolysaccharide-binding protein), with respect to detecting systemic infection and sepsis, according to the Sepsis-3 definition [22]. However, the requirement for measurement using flow cytometry makes the clinical application of CD64 challenging. Firstly, measuring fluorescence intensity requires whole-blood specimens, which raises the potential risk of cell–plasma interaction, which can affect the test results. In addition, the units of flow cytometry measurement for CD64 were not unified. The results could be presented in several forms, including antibody-binding capacity, molecules of equivalent soluble fluorochrome, median fluorescence intensity without standardization, or CD64 index [40]. Most of these results were not mutually convertible due to the requirement for a strictly unified standardization process. Therefore, it is difficult to suggest an optimal cutoff for defining sepsis using CD64. We suggest future studies using CD64 as a marker for diagnosing sepsis present results using the CD64 index [41]. A commercial kit is available for flow cytometry that includes Leuko64 (Trillium Diagnostics, LLC, Brewer, ME, USA). The kit also includes fluorescent beads and antibodies to CD64 and requires lymphocytes and monocytes from patient samples as an internal negative and positive control, respectively [42]. The CD64 index is calculated by taking the ratio of the mean fluorescent intensity (MFI) of cells with CD64 expression to the MFI of beads. The positive control is defined as a CD64 index > 3.0, and the negative control is defined as a CD64 index < 1.0.

### 3.5. Presepsin

Presepsin, a soluble subtype of CD14, is an emerging biomarker of infection and systematic inflammation [43]. Three meta-analysis studies in 2015 investigated the diagnostic value of presepsin for sepsis in the ICU and emergency departments and found pooled sensitivities ranging from 0.77 to 0.86 and specificities ranging from 0.73 to 0.78 [44,45,46]. All three studies concluded that presepsin is a reliable biomarker for diagnosing sepsis. In 2017, we published a systematic review and meta-analysis of the diagnostic accuracy of presepsin in sepsis, which included subgroup analyses comparing to procalcitonin, and CRP. Our study assessed that presepsin had a pooled sensitivity and specificity of 0.84 (95% CI, 0.80–0.87) and 0.76 (95% CI, 0.67–0.82). However, there was no significant difference between presepsin and procalcitonin (AUC 0.87 vs. 0.86) or CRP (AUC 0.85 vs. 0.85) [47]. Brodska et al. conducted a prospective study in 2018 and concluded that presepsin did not outperform procalcitonin and CRP in diagnosing sepsis in critically ill patients [48]. Another systematic review and meta-analysis in 2019 that included 19 studies with 3012 patients also showed a pooled sensitivity and specificity of 0.84 (95% CI, 0.80–0.88) and 0.73 (95% CI, 0.61–0.82) for presepsin and 0.80 (95% CI, 0.75–0.84) and 0.75 (95% CI, 0.67–0.81) for procalcitonin [49]. The author concluded that the diagnostic accuracy of procalcitonin and presepsin in detecting infection are similar and both are therefore useful for early diagnosis of sepsis and subsequent reduction of mortality in critically ill patients. 

Compared with that for procalcitonin, the number of studies reporting the diagnostic value of presepsin for sepsis has only begun to grow significantly in the recent decade. The measurement of presepsin has an instrumental advantage over the measurement of CD64, since it has no definite requirement for whole-blood samples and can provide standardized units for a suggested cutoff [50]. In addition, the final measurement is reported to be available within 1.5 h [51]. Its diagnostic value for sepsis is also promising. Currently, there is insufficient evidence supporting the greater overall diagnostic accuracy of presepsin compared with traditional biomarkers, such as procalcitonin or CRP. However, presepsin may still have an advantage over procalcitonin and CRP for early screening of sepsis [22,47].

### 3.6. Soluble TREM-1 (sTREM-1)

The soluble form of the triggering receptor expressed on myeloid cells-1 (TREM-1), sTREM-1, was first identified by Bouchon et al. in 2000 [52]. Proteolytic cleavage of membrane-bound TREM-1 increases serum sTREM-1 levels and indicates tissue damage by proteinases released by pathogens [53]. A meta-analysis published by Wu and colleagues in 2012 found moderate accuracy for sepsis diagnosis in systemic inflammation patients, which included 11 studies and 1795 patients with a pooled sensitivity of 0.70 (95% CI, 0.65–0.89) and specificity of 0.80 (95% CI, 0.69–0.88) [54]. Another meta-analysis published by Chang et al. in 2020 enrolled 19 studies involving 2418 patients and the pooled sensitivity and specificity were 0.82 (95% CI, 0.73–0.89) and 0.81 (95% CI, 0.75–0.86), respectively [55]. They concluded that sTREM-1 had a moderate ability in diagnosing sepsis. However, the small sample sizes of the enrolled studies, heterogeneities in standard description, prevalence of sepsis, and non-consecution of patient recruitment still limit the strength of the conclusion. In addition, they also found potential publication bias in this meta-analysis.

## 4. Risk of Bias

As identified by the QUDAS-2 criteria and the Newcastle–Ottawa scale, a higher level of evidence that supports clinical utility should include good quality from four domains: patient selection, index test, reference standard, and flow and timing of the study conducted [56,57]. 

In our previous experience of reviewing studies of sepsis biomarkers, few studies adopted random or consecutive sampling strategy, but many studies were conducted using a case-control design. Some studies even tried to use non-infectious SIRS, burn patients, or other situations for which the “extreme” control could be named. The “extreme” case-control would worsen the overestimated performance of biomarkers even more (Figure 2). These flaws may contribute to the heterogeneity and overestimated performance of the biomarkers. The optimistic results obtained by comparing the extremely sick patient to the healthy individual have seldom been translated well in close-to-real-world cohort studies. 

Furthermore, almost all studies attempted to identify the optimal cutoffs of these biomarkers to detect sepsis in a post hoc fashion. The post hoc identification of the cutoff would also be very optimistic given the probability that the cutoff identified in one study would be different from that identified in another study. In other words, researchers were “cherry-picking” the optimal cutoffs in their specific study population, which would again seldom translate into real-world scenarios. Investigators who wish to summarize the overall performance, such as AUC, would also encounter difficulties. In addition, the lack of a gold standard for diagnosing sepsis and small sample sizes are common limitations. Although some researchers have proposed sophisticated statistical methods, such as latent class analysis, results for biomarkers of sepsis are still unavailable [58]. Furthermore, a small sample size, <100 patients, often contributes towards overestimated performance [22], which may account for the less satisfactory results obtained in subsequent studies after the newly discovered biomarkers attract wide attention.

Other than the in-born bias that individual studies possess, we also noticed the inconsistent results generated by the different methods applied in these meta-analyses. For example, Zhongjun Zheng et al. performed a meta-analysis on the accuracy of presepsin for diagnosing sepsis with the Moses–Littenberg SROC method and found that the pooled sensitivity, specificity, and SROC area under the curve were 0.77 (95% CI, 0.75–0.80), 0.73 (95% CI, 0.69–0.77), and 0.8598, respectively [40]. Xin Zhang et al. used the bivariate model for a meta-analysis and found that the pooled sensitivity, specificity, and SROC area under the curve were 0.86 (95% CI, 0.79–0.91), 0.78 (95% CI, 0.68–0.85), and 0.89, respectively [39]. These two meta-analysis studies have only one different enrolled study but reveal pooled results with significant differences, which might be caused by the different methods of meta-analysis used, as the Moses–Littenberg SROC method does not take into account the correlation between sensitivity and specificity and tends to underestimate test accuracy.

Finally, as indicated by one of our systematic reviews and meta-analyses as well as those by many other researchers, sponsorship bias can result in overestimation of biomarker performance. We encourage researchers to routinely examine sponsorship bias in their subgroup analyses or sensitivity analyses [22]. Readers should also examine sponsorship and consider the potential bias this may introduce into studies before implementing the study findings and recommendations into their practices.

## 5. Future Directions

We summarized characteristics of enrolled systematic review and meta-analysis studies in this article in Table 1 and it revealed no single biomarker has outstanding sensitivity and specificity for detecting sepsis and systemic infection at this point; hence, different combinations of biomarkers have been investigated. Kofoed et al. demonstrated that a panel of six biomarkers, including soluble urokinase-type plasminogen activator, sTREM-1, macrophage migration inhibitory factor, CRP, procalcitonin, and neutrophils, more accurately detected patients with bacterial infection than any biomarker alone [59]. However, the model they used can only combine biomarkers linearly; modern machine learning algorithms may manage complicated high dimensionality, non-linear association, and interaction between biomarkers simultaneously [60].

Gibot then demonstrated another combination using CD64, procalcitonin, and sTREM-1, showing good performance with respect to diagnosing sepsis in critically ill patients. They conducted a prospective cohort study of ICU patients that revealed a good diagnostic accuracy of combined biomarkers at 0.95 (95% CI, 0.89–0.99). However, the measurement of CD64 that they included requires flow cytometry, which again prevents clinical utility [61]. As the fundamental concept of sepsis is a life-threatening organ dysfunction caused by a dysregulated host response to infection, the biomarker itself may indicate the host response to infection, but the ability to reflect organ dysfunction is controversial. In Sepsis-3, organ dysfunction is identified by an increase in the Sequential Organ Failure Assessment (SOFA) score of two points or more, thus some researchers advocated combining biomarkers and SOFA score to detect sepsis. Yang et al. developed a scoring system that combined biomarkers, including CRP and procalcitonin, with the SOFA score and found this to be more predictive than any individual marker for diagnosing sepsis in critical patients. The study enrolled only 300 ICU patients and reported the performance of the combined bioscore for diagnosing sepsis with a sensitivity of 0.79, specificity of 0.70, and AUC of 0.790 (95% CI, 0.739–0.834, *p* < 0.001) [62]. Pinar et al. tried to validate the bioscore externally with 226 patients in ICU and found a different sensitivity and specificity (0.46 (95% CI, 0.38–0.53) and 0.85 (95% CI, 0.74–0.93)). However, they enrolled patients with markedly higher 28-day mortality (55.8% instead of 30%), and they adopted a higher bioscore cutoff (4.2 compared to 2.65 in the original article); hence, their validation was unreasonable [63]. Further studies for bioscore components and optimal cutoffs are needed. In addition, the individualized application of biomarkers according to infectious foci, potential pathogen, host comorbidity, or sepsis phenotype is being investigated; however, there is controversy regarding the clinical impact and medicoeconomic effects of biomarkers to date [64,65].

## 6. Conclusions

Based upon the current evidence, we believe CD64 and presepsin should be considered as the most promising biomarkers for diagnosing sepsis. Further studies enrolling a larger sample size and utilizing a cohort rather than a case-control design are warranted. A random or consecutive study design with a pre-specified laboratory threshold, consistent sampling timing, and the updated sepsis definition will also make the studies more reliable. Further investigations of appropriate specimens, testing assays, and cutoff levels for specific biomarkers are also needed. The combination of multiple biomarkers or with clinical scoring systems has better performance than a single biomarker; however, the medicoeconomic effect is still controversial.

## Figures and Tables

**Figure 1 biomedicines-08-00494-f001:**
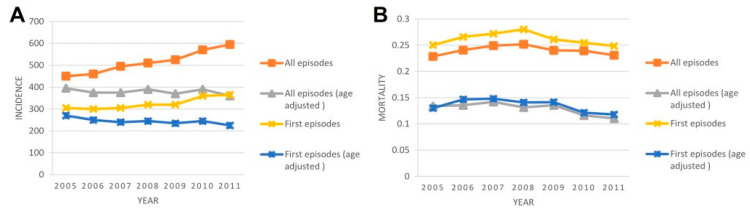
Incidence and mortality of sepsis in Taiwan (incidence presented as event per 100,000 population). (**A**) Incidence trend of severe sepsis. (**B**) In-hospital mortality trend of severe sepsis.

**Figure 2 biomedicines-08-00494-f002:**
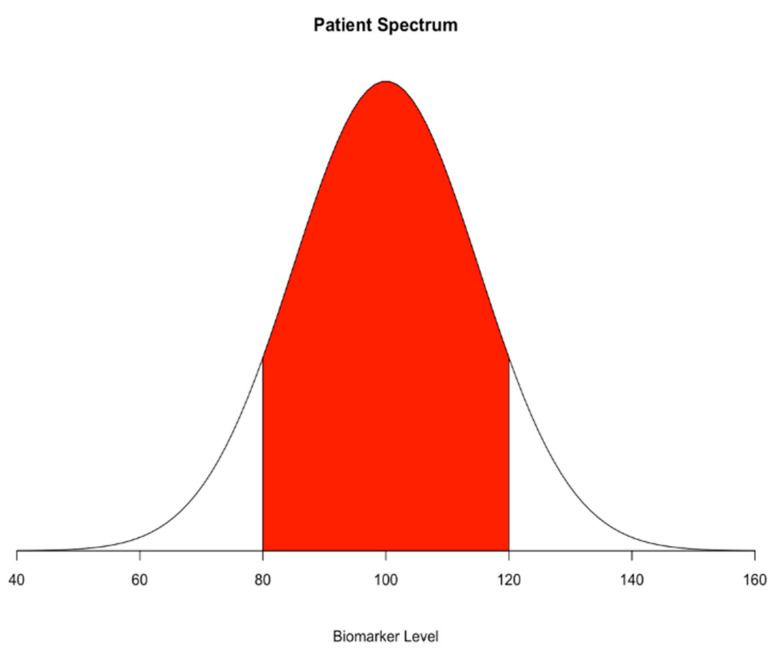
Patient spectrum in case-control studies. Red area indicates the not included population.

**Table 1 biomedicines-08-00494-t001:** Characteristics of enrolled systematic review and meta-analysis studies.

	Assay	Study Inclusion Criteria	Study Exclusion Criteria	Reference Standard	Included Studies/Total Patient Number	Risk of Bias	Outcome (Sensitivity; Specificity; AUROC); Heterogeneity (I2) %	Cutoff	Publication Bias
**CRP**									
Liu 2016	nil	Evaluate the diagnostic accuracy of CRP for distinguishing patients with sepsis from those with non-infectious SIRS	Lacked non-infectious SIRS patients as a control group; Immunocompromised, hematologic and pediatric patients; Not provide sufficient data to build a 2 × 2 contingency table	Culture positive or clinically diagnosed with ACCP/SCCM pre-sepsis 3 definition	45/5654	Most studies were not fulfilled or with unclear representative spectrum of patients	0.75 [0.69, 0.79]; 86.6; 0.67 [0.58, 0.74]; 89.3; 0.77 [0.73, 0.81]	IQR 38–140 mg/L Median 84 mg/L	*p* = 0.71
Tan 2018	nil	English and Chinese article; Clinical trial studies; Adult patients diagnosed with sepsis, severe sepsis, or septic shock in the experience group; noninfectious origin with SIRS in control group; Provide sufficient data to build a 2 × 2 contingency table	Repeat published articles; Data had obvious mistakes; Case report, theoretical research, conference report, systematic review, meta-analysis, expert comment, economic analysis	Clinically diagnosed with ACCP/SCCM pre-sepsis 3 definition	9/1368	No QUADAS assessment	0.80 [0.63, 0.90]; 88.7; 0.61 [0.50, 0.72]; 81.7; 0.73 [0.69, 0.77]	12.00 to 90.00 mg/L	*p* = 0.32
**PCT**									
Wacker 2013	PCT-Q; PCT-Kryptor; PCT-LIA	English, German and French; Differentiate between sepsis patients and SIRS without infection;	Studies that involved healthy people; Studies involving neonates (<28 days); Animal experiments, reviews, correspondences, case reports, expert opinions, editorials	Culture positive or clinically diagnosed with ACCP/SCCM pre-sepsis 3 definition or German Sepsis Society definition	30/3244	Most studies were not fulfilled representative spectrum of patients; Most studies were not fulfilled or with unclear description of reference standard	0.77 [0.72, 0.81]; 77.8; 0.79 [0.74, 0.84]; 78.1; 0.85 [0.81, 0.88]	IQR 0.5-2.0 ng/mL Median 1.1 ng/mL	*p* < 0.0005
**IL-6**									
Ma 2016	ECL; ELISA; CLIA	English articles; Comparing sepsis patients with SIRS without infection; Provide sufficient data to build a 2 × 2 contingency table; Studies including at least 10 patients	Studies involving neonates (<28 days); Animal studies, abstracts, review articles, case reports, letters, editorials, comments, conference proceedings	Culture positive or clinically diagnosed with ACCP/SCCM pre-sepsis 3 definition	22/2680	Most studies with unclear risk of bias for index test, flow and timing	0.68 [0.65, 0.70] 91.6; 0.73 [0.71, 0.76] 77.6; 0.80 [Q*=0.73]	18 to 423.5 pg/mL	*p* = 0.68
Iwase 2019	Roche Diagnostics; BosterBiological Technology; Biosource; Medgenics Diagnostics; DPC Biermann; R&D System	Provide sufficient data to build a 2 × 2 contingency table	Did not investigate the diagnostic accuracy of blood IL-6 level; Animal experiments, case reports, commentaries, letters, meta-analyses, reviews, editorials, meeting abstracts, poster presentations, correspondence	Culture positive or clinically diagnosed with ACCP/SCCM mixed sepsis definition, CDC/NHSN or ISF definition	6/527	Most studies with unclear risk of bias for index test and reference standard	0.73 [0.61, 0.82]; 0.76 [0.61, 0.87]; 0.81 [0.78, 0.85]	35 to 620 pg/mL Median 176 pg/mL	nil
**CD64**									
Wang 2015	FCM; Hematology analyzer; Leuko64 kit	English articles; Provide sufficient data to build a 2 × 2 contingency table	Studies involving neonates (<28 days); Included patients did not have SIRS or were not critically ill	Culture positive or clinically diagnosed with ACCP/SCCM pre sepsis-3 definition	8/1986	QUADAS score between 8-11	0.76 [0.73, 0.78] 92.7; 0.85 [0.82, 0.87] 91.3; 0.95 [Q*=0.89]	nil	*p* = 0.02
Yeh 2019	In-house; Leuko64 kit	English articles Original article; Adult patients;	Duplicated study; Prognosis based on the prediction of mortality from sepsis; Not provide sufficient data to build a 2 × 2 contingency table	Culture positive or clinically diagnosed with ACCP/SCCM pre sepsis-3 definition	14/2471	Most studies with high risk of bias for index test; Most studies with high or unclear risk of bias for patient selection	0.87 [0.80, 0.92] 94.3; 0.89 [0.82, 0.93] 92.0; 0.94 [0.92, 0.96]	nil	*p* = 0.05
**sTREM-1**									
Wu 2012	ELISA; Luminex multiplex assay	Studies assessed the accuracy of plasma sTREM-1 for sepsis diagnosis in adult patients with SIRS; Provided sufficient information to construct a 2 × 2 contingency table;	Studies involving neonates (<28 days); Review article, conference paper, or case report; Did not investigate the diagnostic accuracyof blood sTREM-1 level	Culture positive or clinically diagnosed with ACCP/SCCM pre-sepsis 3 definition	11/1795	Most studies with unclear risk of disease progression bias provided	0.79 [0.65, 0.89] 95.0; 0.80 [0.69, 0.88] 92.7; 0.87 [0.84, 0.89]	40 to 755 pg/mL	*p* = 0.02
Chang 2020	ELISA; Quantitative sandwich enzyme immunoassay; Homemadeenzyme immunosorbent assay; Immunoblots; Luminex multiplex assay; DuoSet enzyme-linked immunosorbent assay	Clinical trials of adult patients (> 18-year-old) with suspected sepsis; Serum or plasma sTREM-1 protein expression; Provide sufficient data to build a 2 × 2 contingency table	Review article, animal study, in vitro study; Prognostic study; Pediatric study;Non-serum sample	Culture positive or clinically diagnosed with ACCP/SCCM pre-sepsis 3 definition	19/2418	Most studies with high or unclear risk of reference standard and patient selection; All studies with unclear risk of index test	0.82 [0.73, 0.89] 93.6; 0.81 [0.75, 0.86] 89.6; 0.88 [0.85, 0.91]	30 to 60,000 pg/mL	*p* = 0.002
**Presepsin**									
Zheng 2015	PATHFAST	Provided the presepsin concentrations of sepsis patients and non-sepsis patients; Provide sufficient data to build a 2 × 2 contingency table;	Reviews, correspondence, editorials, conference abstracts Studies limited to restrictive subgroups	Clinically diagnosed with ACCP/SCCM pre-sepsis 3 definition	8/1757	Most studies with high risk of bias for index test; Most studies with unclear risk of biasfor reference standard, flow and timing	0.77 [0.75, 0.80] 85.2; 0.73 [0.69, 0.77] 80.6; 0.86 [Q* = 0.79]	317 to 729 pg/mL	*p* = 0.755
Xin Zhang 2015	PATHFAST	Comparing sepsis patients with SIRS without infection; Adult patient; Provide sufficient data to build a 2 × 2 contingency table	Reviews, letters, commentaries, correspondence, case reports, conference abstracts, expert opinions, animal experiments; Pediatric study	Clinically diagnosed with ACCP/SCCM pre-sepsis 3 definition	8/1815	Most studies not fulfilled blinding of investigators to index test; All studies were not fulfilled with uninterpretable test results reported	0.86 [0.79, 0.91] 90.5; 0.78 [0.68, 0.85] 91.8; 0.89 [0.86, 0.92]	IQR 317–729 pg/mL Median 560 pg/mL	*p* = 0.31
Jing Zhang 2015	PATHFAST ELISA	Provide sufficient data to build a 2 × 2 contingency table	Duplicate studies; Non-English publications; Conference abstracts; Studies involving asepsis or control sample size <10	Clinically diagnosed with ACCP/SCCM pre-sepsis 3 definition, ABA, IPSCG or ISF definition	11/3052	Most studies with high or unclear risk of bias for patient selection and index test	0.83 [0.77, 0.88] 84.3; 0.78 [0.72, 0.83] 86.0; 0.88 [0.84, 0.90]	317 to 729 pg/mL	*p* = 0.12
Wu 2017	PATHFAST	English articles; Sepsis related studies including Diagnostic studies	Non-sepsis related studies; Non-diagnostic studies; Studies with no performance parameters given; Non-original studies; Non-blood specimen	Culture positive or clinically diagnosed with ACCP/SCCM mixed sepsis definition, ABA, or SEIMC definition	18/3470	All studies with high risk of index test	0.84 [0.80, 0.87] 82.0; 0.76 [0.67, 0.82] 90.2; 0.88 [0.85, 0.90]	IQR 439–664 pg/mL Median 600 pg/mL	*p* = 0.68
Kondo 2019		Sepsis 3, severe sepsis or septic shock with Sepsis 1,2 definition; Cross-sectional, cohort, case-control and randomized controlled trials; Plasma or serum study	Predominantly comprising neonates or perioperative patients; Comprising healthy participants as controls; Not provide sufficient data to build a 2 × 2 contingency table; Animal study	Culture positive or clinically diagnosed with ACCP/SCCM mixed sepsis definition	19/3012	Most studies with high or unclear risk of bias for index test; Most studies with unclear risk of reference standard	0.84 [0.80, 0.88] 62.4; 0.73 [0.61, 0.82] 86.7; 0.87 [0.84, 0.90]	106.1–907 pg/mL	*p* = 0.35

CRP: C-reactive protein; SIRS: systemic inflammatory response syndrome; ACCP/SCCM: American College of Chest Physicians/ Society of Critical Care Medicine; PCT: procalcitonin; ECL: electrochemiluminescence; ELISA: enzyme-linked immunosorbent assay; CLIA: chemiluminescence immunoassay; CDC/NHSN: Center of Disease Control and Prevention/National Healthcare Safety Network; FCM: flow cytometry; IQR: interquartile range. ISF: International Sepsis Forum; IPSCG: International Pediatric Sepsis Consensus Guidelines; ABA: American Burn Association. SEIMC: Spanish Society of Infectious Diseases and Clinical Microbiology.ACCP/SCCM pre sepsis 3 definition: include ACCP/SCCM sepsis 1 and 2 definition; ACCP/SCCM Mixed sepsis definition: include ACCP/SCCM sepsis 1, 2 and 3 definition.

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
