# Peer review of "Current Evidence and Limitation of Biomarkers for Detecting Sepsis and Systemic Infection"

_biomedicines, 2020, doi:10.3390/biomedicines8110494_

Round 1

Reviewer 1 Report

In this narrative review, six promising biomarkers used to diagnose sepsis are discussed, and the authors for several reasons find CD64 and presepsin as the most promising ones.

This is a very subjective review, and no formal jugdement about the best ones are done, but the autjors have sound discussions and refer to recent studies within each biomarker. As such it is a good overview of the present situation.

My major concern is that these biomarkers will probably never be able to set the diagnosis of sepsis on their own. As they write, the new Sepsis 3 criteria from 2016 defines sepsis as 1) strong indications of an infectious process AND 2) new acute organ dysfunction. In my opinion biomarkers reflecting the inflammation process and body response can at present only help with the first part, the last one, organ dysfunction must be solved with the help of other tools, like an organ dysfunction score like the SOFA score. Hence already in he title this goes in way wrong, since the diagnosis is based on two equally important presentations. 

I miss this in the discussion and urge the author to take this up in a revised manuscript.

Minor Issue: Sepsis 1 was abandoned long before 2016, and it was the sepsis 2 definition with organ failure suggestions from 2001 (ref CCM 2003; 31: 1250-6) that was abandoned in 2016 with the Sepsis 3 definitions.

Author Response

Major comments: 

- My major concern is that these biomarkers will probably never be able to set the diagnosis of sepsis on their own. As they write, the new Sepsis 3 criteria from 2016 defines sepsis as 1) strong indications of an infectious process AND 2) new acute organ dysfunction. In my opinion biomarkers reflecting the inflammation process and body response can at present only help with the first part, the last one, organ dysfunction must be solved with the help of other tools, like an organ dysfunction score like the SOFA score. Hence already in the title this goes in way wrong, since the diagnosis is based on two equally important presentations.

Authors response:  

We appreciated the point that solely biomarker may not enough to diagnose sepsis, we revised our title from “Current Evidence and Limitation of Biomarkers for Diagnosing Sepsis” to “Current Evidence and Limitation of Biomarkers for Detecting Sepsis and Systemic Infection”, we also revised the concept in the manuscript (line 18-19; 299-300). Besides, we added a paragraph from line 311-316 to discuss investigators combining biomarkers and organ dysfunction score to detect sepsis.

“The fundamental concept of sepsis is a life-threatening organ dysfunction caused by a dysregulated host response to infection, the biomarker itself may indicate the host response to infection but the ability to reflect organ dysfunction is controversial. In Sepsis-3, organ dysfunction is identified by an increase in the Sequential Organ Failure Assessment (SOFA) score of 2 points or more, thus some researchers advocated combining biomarkers and SOFA score to detect sepsis.”

Minor comments:

- Sepsis 1 was abandoned long before 2016, and it was the sepsis 2 definition with organ failure suggestions from 2001 that was abandoned in 2016 with the Sepsis 3 definitions.

Authors response:

We revised the manuscript from line 36-38.

“The conventional definition was abandoned in 2016, and "sepsis 3-0" replaced the previous "severe sepsis" to increase the predictive accuracy.”

Reviewer 2 Report

The Authors summarized very exactly the role of potential biomarkers in sepsis. There are only two critical theoretical comments.

  1. In chapter of introduction the pre sepsis-3 and after sepsis-3 period is well characterized, because the sepsis definiton extremely changed. Thus, in different chapters the diagnostic accuracy of different biomarkers depends on different sepsis conditions. How may be solved this problem?  I suggest to note for every citation, whether the source is using data from pre sepsis-3, after sepsis-3, or mixed populations. The date of publication itself may not help, some publications in 2018 are using the pre sepsis-3 definition.
  2. The senitivity and specificity normally can be determined at an exactly defined cutoff value. If possible, these cutoff values should be published as well. In simple cohorts it is well characterized, some metaanalysises contain median values (e.g. citation 25., Wacker et al. - median value 1.1 ng/ml). Regarding the CD64 due to the different incomparable methods the inclusion of cutoffs is not mandatory.

Author Response

Major comments:

  • In chapter of introduction the pre sepsis-3 and after sepsis-3 period is well characterized,

because the sepsis definition extremely changed. Thus, in different chapters the diagnostic accuracy of different biomarkers depends on different sepsis conditions. How may be solved this problem?  I suggest to note for every citation, whether the source is using data from pre sepsis-3, after sepsis-3, or mixed populations. The date of publication itself may not help, some publications in 2018 are using the pre sepsis-3 definition.

  • The sensitivity and specificity normally can be determined at an exactly defined cutoff value.

If possible, these cutoff values should be published as well. In simple cohorts it is well characterized, some meta-analyses contain median values (e.g. citation 25., Wacker et al. - median value 1.1 ng/ml). Regarding the CD64 due to the different incomparable methods the inclusion of cutoffs is not mandatory.

Authors response:

To assist readers understanding enrolled systematic review and meta-analysis studies in this article, we summarized a table of characteristics for each biomarker (Table1).  In the column of reference standard, we categorized ACCP/SCCM with pre-sepsis 3(sepsis 1,2) and mixed definition (sepsis 1,2,3). We also summarized cutoff value in another column.

Round 2

Reviewer 1 Report

I am satisfied with the authors response to my suggestions for change and making the distinction between inflammatory response and organ dysfunction better.

This manuscript is a resubmission of an earlier submission. The following is a list of the peer review reports and author responses from that submission.